Novel Systems Biology Techniques

# High-Throughput Stool Metaproteomics: Method and Application to Human Specimens

Carlos G. Gonzalez,[a] Hannah C. Wastyk,[b] Madeline Topf,[c] Christopher D. Gardner,[d] Justin L. Sonnenburg,[c] Joshua E. Elias[e]

[a]Department of Chemical and Systems Biology, Stanford School of Medicine, Stanford University, Stanford, California, USA
[b]Department of Bioengineering, Stanford University, Stanford, California, USA
[c]Department of Microbiology and Immunology, Stanford School of Medicine, Stanford University, Stanford, California, USA
[d]Stanford Prevention Research Center, Department of Medicine, Stanford School of Medicine, Stanford, California, USA
[e]Chan Zuckerberg Biohub, San Francisco, California, USA

**ABSTRACT** Stool-based proteomics is capable of significantly augmenting our understanding of host-gut microbe interactions. However, compared to competing technologies, such as metagenomics and 16S rRNA sequencing, it is underutilized due to its low throughput and the negative impact sample contaminants can have on highly sensitive mass spectrometry equipment. Here, we present a new stool proteomic processing pipeline that addresses these shortcomings in a highly reproducible and quantitative manner. Using this method, 290 samples from a dietary intervention study were processed in approximately 1.5 weeks, largely done by a single researcher. These data indicated a subtle but distinct monotonic increase in the number of significantly altered proteins between study participants on fiber- or fermented food-enriched diets. Lastly, we were able to classify study participants based on their diet-altered proteomic profiles and demonstrated that classification accuracies of up to 89% could be achieved by increasing the number of subjects considered. Taken together, this study represents the first high-throughput proteomic method for processing stool samples in a technically reproducible manner and has the potential to elevate stool-based proteomics as an essential tool for profiling host-gut microbiome interactions in a clinical setting.

**IMPORTANCE** Widely available technologies based on DNA sequencing have been used to describe the kinds of microbes that might correlate with health and disease. However, mechanistic insights might be best achieved through careful study of the dynamic proteins at the interface between the foods we eat, our microbes, and ourselves. Mass spectrometry-based proteomics has the potential to revolutionize our understanding of this complex system, but its application to clinical studies has been hampered by low-throughput and laborious experimentation pipelines. In response, we developed SHT-Pro, the first high-throughput pipeline designed to rapidly handle large stool sample sets. With it, a single researcher can process over one hundred stool samples per week for mass spectrometry analysis, conservatively approximately $10\times$ to $100\times$ faster than previous methods, depending on whether isobaric labeling is used or not. Since SHT-Pro is fairly simple to implement using commercially available reagents, it should be easily adaptable to large-scale clinical studies.

**KEYWORDS** diet, fermented, fiber, high-throughput, mass spectrometry, metaproteomics, microbiome, proteomics, stool

The gut microbiome is characterized by numerous complex interactions, influencing human health and disease, and has been associated with disorders ranging from inflammatory bowel disease to autism (1, 2). Stool is a biologically rich biomaterial,

Address correspondence to Joshua E. Elias, josh.elias@czbiohub.org.

SHT-Pro—a fast metaproteomics method from the Elias lab could help discover how diet and the gut microbiome impact human health from large clinical studies.

containing host, microbe, and dietary proteins, among a rich array of biomolecules. The broad proteinaceous representation of relevant biological entities and interactions, in conjunction with noninvasive sample collection, makes stool ideal for studying the complex ecosystem at the host-gut microbe interface (3). Microbiome composition can be readily determined using 16S rRNA gene sequencing from stool DNA and, due to its high-throughput nature, is well-suited for surveying a single individual over extended longitudinal time courses. Metagenomic and metatranscriptomic sequencing technologies can elucidate microbes' functional capacities and states (4). However, additional measurements are needed to elucidate the microbiome-host interactions that can profoundly affect host health.

Stool proteomics offers the ability to simultaneously measure both host- and microbe-expressed proteins, their posttranslational modifications, and the dietary components also present in the gut. These components reflect interactions and physiological states that are otherwise difficult to survey through nucleic acid sequencing alone (5). We previously showed that host proteins in stool reflect expression along the length of the gut and reveal signatures specific to the type of inflammatory state, such as distinct levels of antimicrobial proteins. Importantly, these signatures can vary in a manner distinct from that of the gut microbiota (5). For example, we showed previously that fecal microbiota transplanted into an antibiotic-induced *Clostridium difficile* infection mouse model normalized the microbial composition but not the host stool proteomic profile. Since proteins can be recovered from archived frozen stool samples, the approach offers a way to illuminate aspects of host mucosal biology noninvasively and longitudinally, long after stool collection.

Despite its functional utility, stool-based metaproteomics remains underutilized compared to the aforementioned next-generation sequencing technologies. One major hindrance to broader implementation has been low sample processing throughput. Indeed, while we and others previously demonstrated the power and utility of stool proteomics, those studies relied on data generated at rates as low as 10 to 30 samples per week (6–9). Additionally, workflows developed for processing cell culture and tissue lysates are not optimized to eliminate contaminating molecules that are abundant in stool. Insufficient contaminant removal can lead to instrumentation downtime, decreases overall sample throughput, and introduces experimental noise that can dilute biologically relevant signals.

Here, we describe a method, the Stool High-Throughput Proteomics pipeline (SHT-Pro), that increases our ability to acquire high-quality metaproteomic stool analyses by as much as 100-fold when paired with multiplexing technologies such as tandem mass tag (TMT) labeling. As a first demonstration of this method, we applied SHT-Pro to 145 stool specimens longitudinally collected from 29 human participants as part of an ongoing dietary intervention study investigating the biological effects of diets enriched in fiber versus fermented foods (ClinicalTrials.gov registration no. NCT03275662). Processing them in duplicate (290 total samples) using SHT-Pro took approximately 1 week from stool to mass spectrometry-ready peptides, an estimated time savings of over 2.5 months compared to our previously published workflow (5, 6). The resulting data set identified over 5,600 unique host and microbial proteins, 45% of which were shared between both study groups. We found that the number of proteins that significantly differed between the two groups increased over time, indicating that diet shapes the stool metaproteome of humans. We further demonstrate that the inclusion of more participants in metaproteomic analyses, in a fashion that this method enables, enhances the ability to classify study subjects compared to the smaller-scale data sets that were more feasible using prior methods. These data support SHT-Pro as overcoming a major hindrance for performing the kinds of clinical-scale studies needed for statistically sound measurements of diet and its impact on the host and its gut microbiome.

## RESULTS

**SHT-Pro increases sample processing speed with a high degree of reproducibility.** A major limitation to large-scale adoption of stool metaproteomics has been its

heavily reduced throughput compared to that of DNA sequencing. Considering the labor-intensive, multiday nature of our previously published stool proteomics method, we found that one researcher could reasonably process 25 stool samples per week on average (6, 8). To narrow this gap, we developed our pipeline to rapidly process hundreds of stool samples (Fig. 1A) in a matter of days while maximizing liquid chromatography and mass spectrometry (LC-MS) instrumentation stability. Ninety-six-well protein trap columns (Protifi S-trap) are robust to a wide range of protein/trypsin ratios, making them suitable for stool specimens with various protein contents. Using them for initial protein purification and digestion along with automation technologies for solid-phase extraction cleanup are two critical components of this added efficiency. To test time savings of the new method, we compared sample processing time of our previously published workflow to that of SHT-Pro (Fig. 1B). While processing fewer than 10 samples at a time does not result in substantial time savings (~2.5 to 3 days saved), larger sample sets benefit from dramatic time savings. For example, processing 96 samples (a single 96-well plate) takes as little as 1.5 days using SHT-Pro, compared to approximately 30 days using the previous protocol (approximately 20-fold decrease).

The sample processing speed improvements SHT-Pro provides would be of little value without effective contaminant removal. To evaluate the kind of contamination-dependent analytical degradation that can occur over time, we repeatedly injected a single SHT-Pro processed stool specimen into our mass spectrometer 20 times. Four analyses of a standard complex peptide mixture were interspersed throughout these stool LC-MS analyses: one prior to all stool LC-MS analyses, two spaced 10 stool analyses apart, and one following 20 stool analyses. We observed no substantial degradation of LC-MS performance as measured from search results of the standard peptide mixture ($7,205 \pm 60$ unique peptides) versus four LC-MS analyses of the standard mixture on a new analytical column (average of $7,350 \pm 150$ unique peptides). In contrast, we observed a 30% decrease in peptide spectral matches (PSMs) and a 27% decrease in peptide identifications in our standard peptide mix using our previous method over a similar number of injections ($n = 16$ injections) (see Fig. S1A in the supplemental material). While sample purity and mass spectrometer performance are also responsive to other factors, such as desalting protocols, the amount of sample loaded onto the column, and instrument type, these results suggest that peptides resulting from the SHT-Pro pipeline are not substantially contaminated in a way that impairs sensitive LC-MS equipment.

We next tested whether our preparation method also led to high reproducibility. To accomplish this, we aliquoted the stool specimen described above in various amounts (50, 100, and 200 mg) and processed each aliquot by SHT-Pro in duplicate (Fig. S1C). Starting material amounts were chosen based on our previous protocol as well as what we have found to be generally available from human clinical studies. We note, however, that we did not test the lower limit of initial starting material needed for SHT-Pro or attempt to control for the large amount of variation found in stool sample consistency. These analyses identified 11,373 unique peptides originating from 1,879 (1,152 microbial, 727 host) proteins, averaging 1,791 proteins per sample. Over 85% of all proteins were identified from all preparative replicates and starting material amounts, suggesting a low degree of sample preparation bias (Fig. S2A). All input protein amounts produced strong linear correlations ($R^2$ values for 50 mg = 0.85, 100 mg = 0.92, and 200 mg = 0.91) (Fig. 1E), suggesting that approximately 100 mg of starting material is sufficient for technical reproducibility. Similarly, comparing the intensity of proteins found in a replicate of the 50-mg samples to those of the 200-mg samples yielded an $R^2$ value of 0.86 (Fig. S2B). As expected, these preparative replication correlation values were less than correlations between technical replicate LC-MS sampling from the same SHT-Pro-prepared peptide mixture ($R^2 = 0.99 \pm 0.001$, $n = 6$ pairs). These data suggest a high degree of sample-to-sample processing fidelity.

We next examined how new procedural components of SHT-Pro compared to the lower-throughput aspects of our previously published workflow, specifically bead beating versus vortexing and S-trap protein isolation versus trichloroacetic acid (TCA)

   

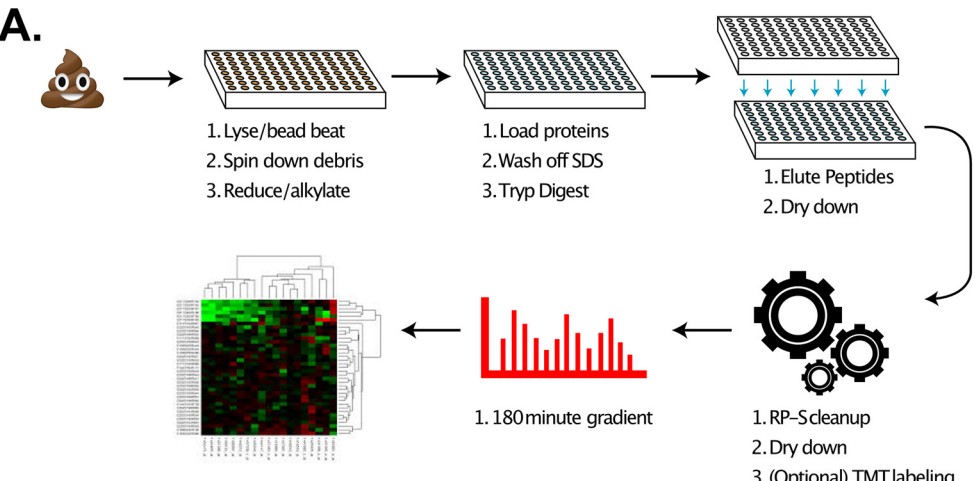

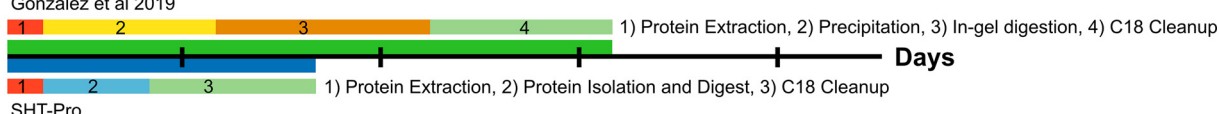

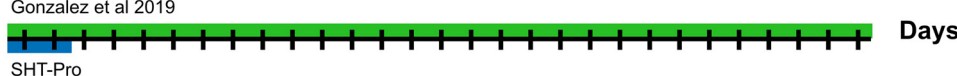

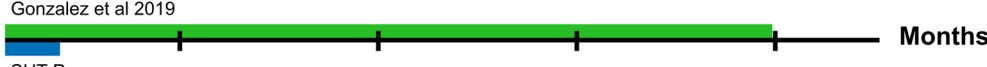

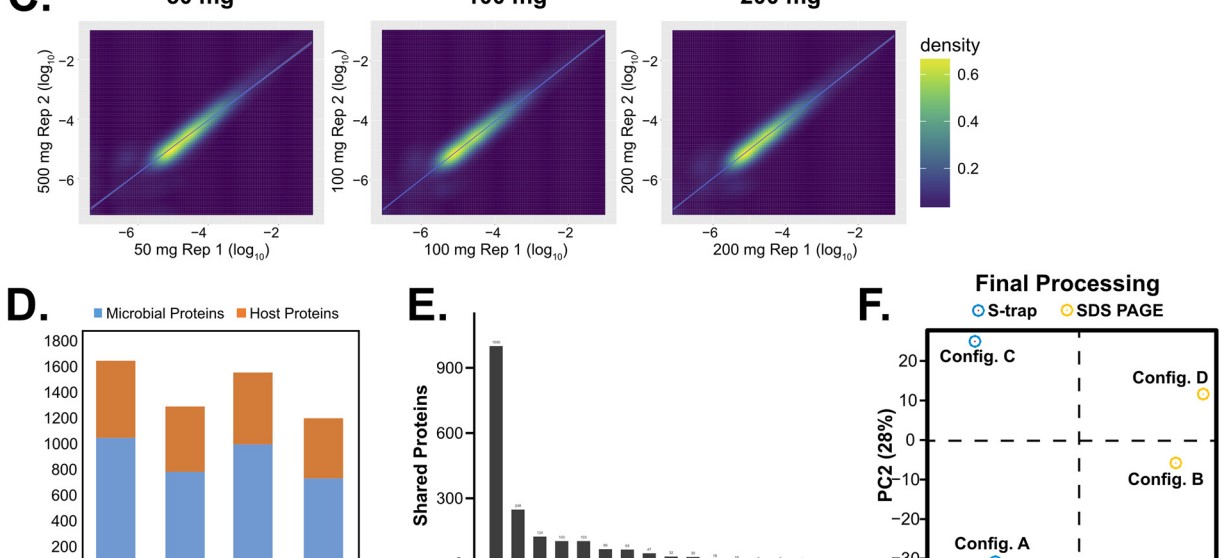

**FIG 1** SHT-Pro is highly reproducible and reveals biologically relevant information. (A) Simplified SHT-Pro processing pipeline. (B) A comparison of estimated time taken to process various amounts of samples. Multicolored bars in the first section represent general pipeline stages. (C) A scatterplot

**TABLE 1** Configuration of samples used in comparison of SHT-Pro and previous workflow[a]

| Sample | Bead-beating | Vortex | TCA | S-trap | SDS-PAGE |
| --- | --- | --- | --- | --- | --- |
| Config. A | X | | | X | |
| Config. B | X | | X | | X |
| Config. C | | X | X | X | |
| Config. D | | X | X | | X |

[a]The table outlines the preparation conditions for each sample.

precipitation combined with SDS-PAGE. We evaluated the number of overall identifications made with two variations of each method, using parallel aliquots of the same stool sample described in Fig. 1A (Table 1 describes sample configurations) (Fig. S1D). Combined, the four measurements yielded 2,352 total protein identifications. Both samples processed with S-trap protein isolation and digestion had numbers of protein identifications similar to those of the SHT-Pro pilot detailed above (average of 1,610 ± 66). Samples processed with TCA precipitation and SDS-PAGE purification yielded fewer protein identifications (average of 1,255 ± 65). Of note, samples processed with the complete SHT-Pro yielded the greatest number of identifications (1,657) (Fig. 1D). Comparing proteins found in this SHT-Pro sample to those from the initial pilot yielded an $R^2$ value of 0.67 ± 0.014 despite originating from two different preparative replicates thawed and processed months apart.

The ratios of microbe-to-host protein identifications were slightly higher for SHT-Pro-prepared samples (mean, 1.8) than for our previous workflow (mean, 1.5). Despite this larger proportion of microbe protein identifications, we found that the numbers of host proteins identified only with SHT-Pro (595, config. A) were substantially greater than the number of host proteins identified with our previous workflow (467, config. B). As with the experiment described in Fig. S1C, the largest subset of proteins (1,000, 43% of all proteins) was present in all samples, regardless of preparation pipeline (Fig. 1E). The next largest unique protein set (248) included those shared only by the two S-Trap-prepared samples (config. A and C) and not identified in the SDS-PAGE preparations (config. B and D). Only 32 proteins were found solely in SDS-PAGE-prepared samples (B and D). We attribute the decreased overlap (43% versus 85%) of this data set compared to the dilution-series sample set (Fig. S2A) to the differing sample preparation conditions of each experiment. The two samples that included SDS-PAGE were more alike in their proteomic profile than the two samples that were processed with the S-Trap (config. A and C) (Fig. 1F). This is likely due to config. C's use of TCA precipitation prior to S-trap processing while config. A did not, which may cause an increase in specific protein subsets.

Given the improved speed, reproducibility, and sensitivity we observed with SHT-Pro, we next tested whether this method tended to identify proteins with biological relevance to the gut environment. We subjected the 100 most abundant proteins to gene ontology enrichment analysis using ShinyGO (10). This revealed that the source stool specimen was significantly (false discovery rate [FDR] of <0.05) enriched for antimicrobial activity, neutrophil activation markers, and increased protease activity (Fig. S3). Given that the specimen set originated from a patient with inflammatory bowel disease (IBD) during a flare, this result agrees with observations we (11) and others (12) have previously made and supports SHT-Pro's ability to produce biologically relevant information.

**FIG 1** Legend (Continued)
comparison of two replicate SHT-Pro analyses of a single stool specimen collected from an IBD patient during flare, using differing amounts of starting material but the same amount of sample loaded onto our chromatography columns (0.5 μg). (D) Comparison of identified proteins from four different conditions: SHT-Pro (config. A); bead beating, TCA precipitation, and SDS-PAGE purification (config. B); vortex only, TCA precipitation, and SHT-Pro (config. C); or previous workflow (config. D). Each bar represents the number of identified microbe or host proteins. See Table 1 for full experimental conditions. (E) Shared subset plot of all four samples. (F) PCA comparing the four SHT-Pro and previous workflow samples described in Fig. S1B. Protein abundances were normalized and scaled (log$_2$) prior to analysis.

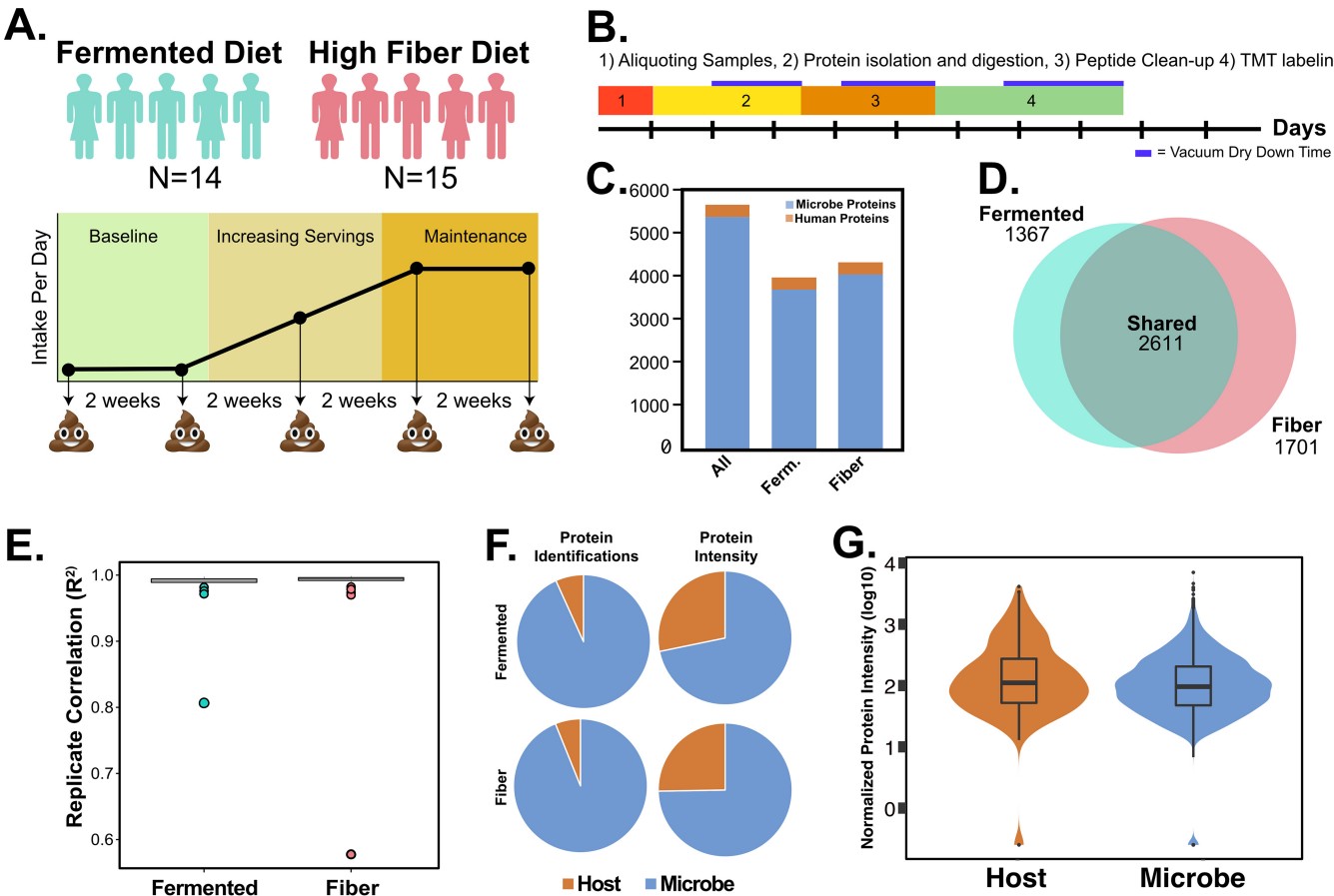

**FIG 2** Diet study design and result overview. (A) Illustration of major diet study components. (B) Illustration of time taken for each step of SHT-Pro during the fiber and fermented group test. Section one and part of section two were done by three researchers, while sections three and four were done by a single researcher. (C) Bar graph of overall number of proteins identified as well as proteins identified within each diet group. (D) Box plot of $R^2$ values for fiber and fermented groups (normalized and $\log_2$-transformed values). Average $R^2$ value for fermented group was 0.997, while that for the fiber group was 0.996. (E) Venn diagram of shared and unique proteins identified in diet groups. (F) Pie charts comparing protein identifications per group and organism (host or microbe) as well as percent intensity of microbes and host proteins. (G) Violin plot comparison of host protein intensity/microbial protein intensity using various scales. Data are $\log_{10}$ transformed.

**Application of SHT-Pro to a longitudinal human diet study.** The advances in throughput, coupled with high reproducibility and ability to reveal biologically relevant gastrointestinal response pathways, make SHT-Pro amenable to large sample sets that were previously impractical to process. To demonstrate SHT-Pro's utility on a large, longitudinal data set, we applied it to samples collected from an ongoing dietary intervention study (ClinicalTrials.gov registration no. NCT03275662). The overarching goal of this study is to elucidate how diets enriched in high fiber (e.g., whole grains, legumes, and fruits) or in fermented (e.g., kombucha, kimchee, and yogurt) foods affect human health. Over the course of 4 months, study participants increased their intake of one of the two dietary intervention arms, and stool specimens were collected every 2 weeks over four phases: baseline, ramp-up (increasing intake), maintenance (peak intake), and choice (choosing to eat the respective diet or not). We selected a subset of patients ($n = 29$) for metaproteomic analysis based on sample availability during the baseline (two samples), ramp-up (single sample), and maintenance (two samples) phases for a total of five samples from each person over this period. The resulting 145 stool specimens were processed in duplicate, for a total number of 290 stool measurements (Fig. 2A). Digested peptides resulting from initial processing with SHT-Pro were chemically labeled with tandem mass tag multiplexing (TMT-11plex) labels to increase throughput and quantifiability. Each TMT-11plex set contained one subject's full time

course in duplicate (5 time points $\times$ 2 replicates) plus one bridge channel representing a mixture of all 290 samples collected.

Once all 290 stool samples were transferred to four 96-well plates, they were processed over the course of <9 days, including approximately 2.5 days devoted to TMT labeling and the associated cleanup steps that follow labeling. A single researcher carried out 80% of these steps (Fig. 2B). This sample set resulted in the identification of 83,061 high-confidence ($q < 0.01$) peptides (16,463 unique) assigned to 5,679 protein families (Fig. 2C and Tables S1 and S2). Of these, approximately 94% (5,372) of identified proteins originated from microbes, with a much smaller host protein set (307). We found a large group of proteins (2,611, 46%) was shared by participants in both groups, while 54% (3062) of proteins identified were uniquely identified within just one dietary subgroup (fermented, 1,361; fiber, 1,701) (Fig. 2D). Replicate stool preparations were highly correlated (average $R^2$ of >0.995 for both groups) with only 2 of 145 replicate pairs receiving an $R^2$ value of <0.90 (0.81 and 0.57), confirming a high degree of overall preparative reproducibility (Fig. 2E).

Having established the stability of SHT-Pro, we next focused on how microbial and host proteins compositionally contributed to the data set at a high level. Despite comprising just 8% of all proteins identified, host-expressed proteins claimed much larger proportions of overall protein abundances (fermented , 25%; fiber, 28%) with an average host protein intensity approximately 67% (fermented, 72%; fiber, 63%) greater than those of their microbial counterparts (Fig. 2F and G). At the level of individual study participants, the fermented group had an average of 805 proteins in each sample, while the fiber group had 870 proteins (Fig. S4C). This difference was not significant ($P = 0.19$ by unpaired $t$ test), suggesting that both groups identified similar numbers of proteins despite the differences in diet. Together, these data demonstrate SHT-Pro workflow yields quantitatively consistent metaproteomic measurements when used with TMT labels or label-free quantification.

**SHT-Pro highlights presence of diet-responding proteomic subset.** Having shown the efficacy of SHT-Pro in generating large and reproducible metaproteomic surveys, we next sought to understand if these two diets had any discernible effects on the stool proteome. Comparing microbe and host-expressed proteins via principal component analysis (PCA) (Fig. S5A) suggested several global trends. First, we found that microbial protein variation across all study participants was largely explained by the first principle component (37%). Three participants within the fiber group were distinguished from the other participants by PC2. In contrast, host proteins exhibited less subject-specific clustering. Overall, neither microbial nor host protein measurements could clearly distinguish diet-induced effects at this high-dimensional level, whereas individual-specific microbial protein expression was much more substantial. This observation aligns with previous reports of microbiome composition profiles measured via 16S rRNA amplicon sequencing (13, 14).

Having observed minimal intergroup differences from high-level analysis, we next sought to determine whether biologically relevant temporal trends could be deduced at a more granular level. Comparing the proteomes of the two diet groups at each time point, we detected a trend suggesting diverging expression of both host and microbial proteomes subsequent to diet augmentation (Fig. 3A and Fig. S6B). More specifically, we observed an increase in significantly altered host (ramp, 9; maintenance, 17) and microbial (ramp, 10; maintenance, 45) proteins subsequent to the start of diet augmentations, although the significance of these small-number observations did not always surpass strict (FDR < 0.05) multiple-hypothesis testing (Fig. S6B and C). Nevertheless, the majority of these host proteins (14/17) increased among fermented group participants while exhibiting negligible overall change in the fiber group. The STRING protein-protein interaction and Gene Ontology (GO) platform suggested these 15 proteins were enriched (FDR < 0.05) in GO terms, including maintenance of intestinal epithelium, glycosylphosphatidylinositol anchor binding, and sphingolipid metabolism (Fig. 3B). It is notable that 9 of these 17 host proteins were also among the subset of

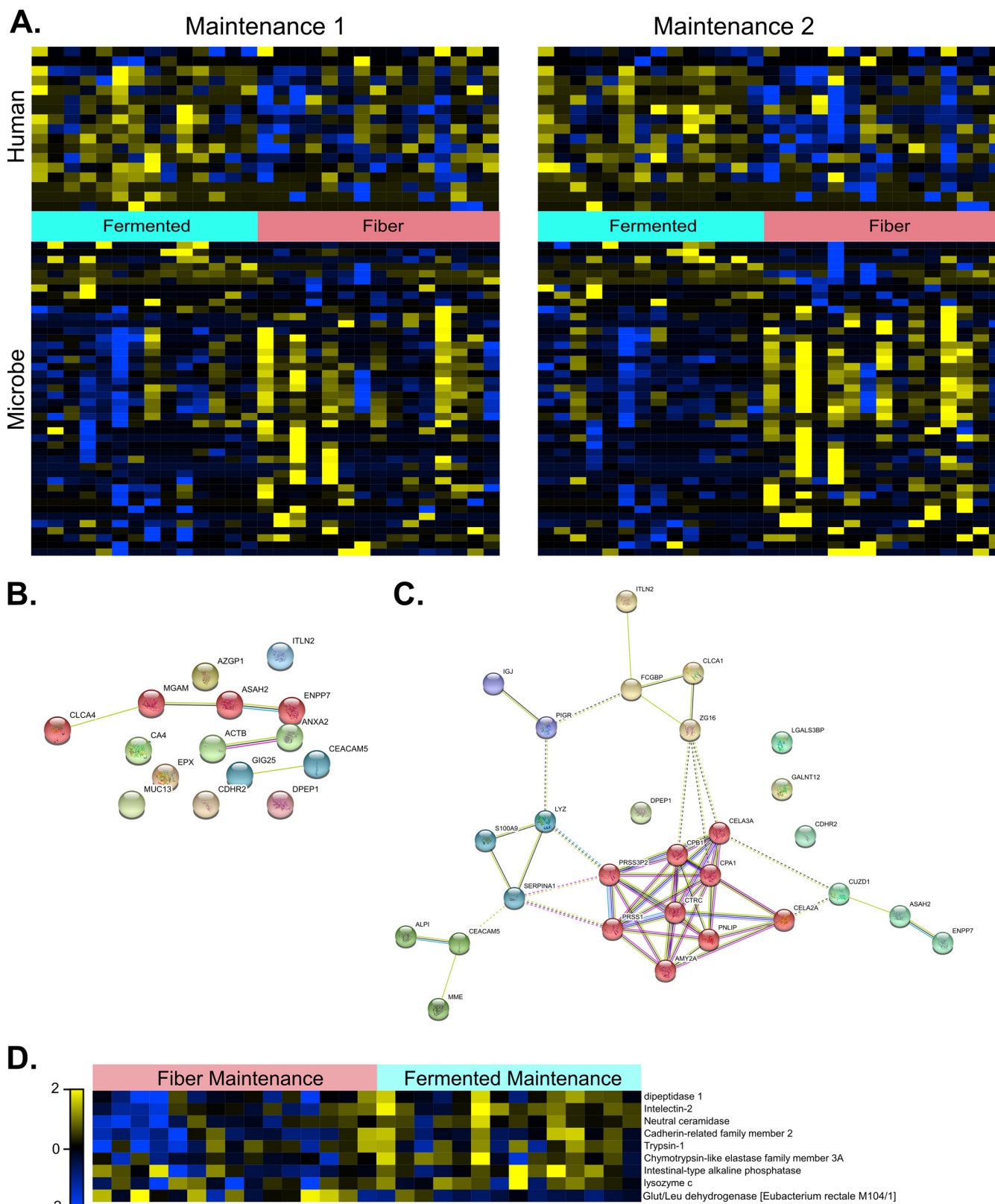

**FIG 3** Diet intervention proteome reveals both unique and core proteomes. (A) Proteins that were significantly altered ($P < 0.05$) between fermented and fiber groups during the maintenance phase (time point 1 or 2). Proteins were also filtered for variation between participants ($\sigma/\sigma_{max\ variation}$, $>0.15$). See Table S3 for an accompanying list of significantly altered proteins and their normalized abundances. (B) StringDB-generated functional network map using proteins significantly increased in the fermented group at the final maintenance time point. Nodes are colored according to the result of Markov-Clustering algorithm employed by StringDB, with each color signifying unique functional subnetworks. (C) StringDB-generated functional network map of the commonly shared

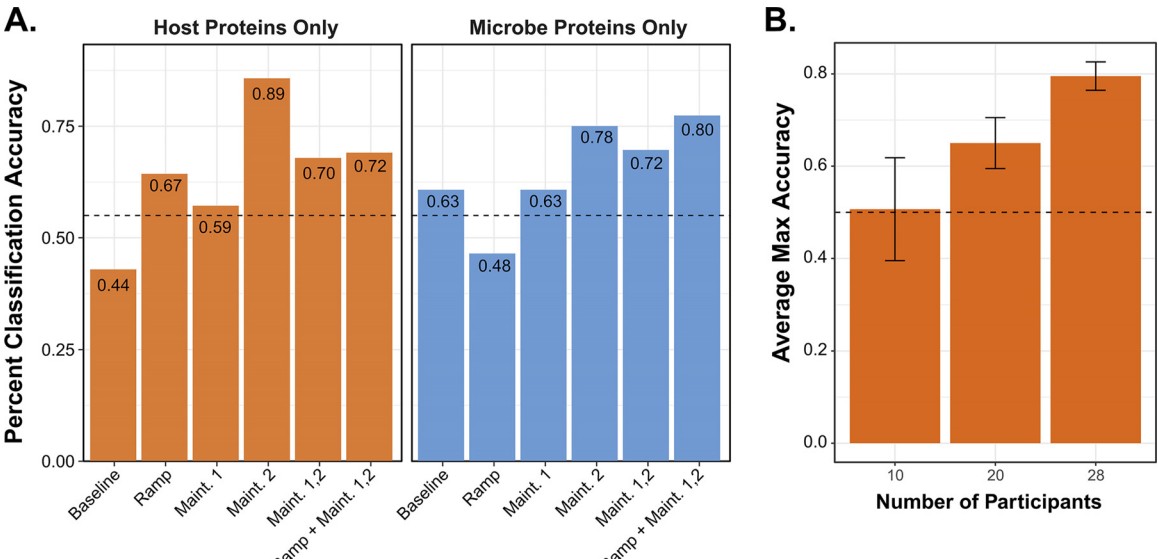

**FIG 4** SHT-Pro-generated metaproteomes classify diet study participants. (A) Bar plot highlighting classification accuracy of the random forest model using either microbial or host proteins from various combinations of points along the time course depicted in Fig. 2A. Data were normalized by sample intensity and scaled to protein intensities found on the first baseline day. (B) Bar plot comparing the LOOCV random forest model's average maximum classification accuracy (e.g., average recorded maximum accuracy for each round of the leave-one-out classification scheme) using only host proteins and varying the number of participants considered ($n = 10$, 20, and 28). Ramp and both maintenance phase days were used for each model, while the participant number was varied.

33 proteins identified in all study participants, regardless of diet group (Fig. 3D), and therefore could be useful markers of a wide range of host responses. As expected, the protein set common to all participants was strongly enriched (FDR < 0.05) in GO terms commonly found in the gut (Fig. 3C and Table S4). These results suggest diet augmentation with fiber or fermented foods has a distinguishable impact on both host and microbial proteomes and highlights their ability to affect the expression of highly prevalent gut-related proteins.

These results indicate the stool proteome can be conceptually divided into several subset proteomes: an individual-specific proteome largely made of microbial proteins, a diet-impacted proteome, and a common core proteome functionally associated with digestion and largely made up of host proteins common to most individuals. Given both common and unique proteome sets can be readily measured from all subjects, we next focused on whether these proteomes could be used to predict membership in either the fermented or fiber protein groups.

**SHT-Pro-derived protein abundance allows for classification based on diet group.** As indicated in Fig. 3, we observed modest diet-related differences between the fermented and fiber groups. However, we were curious as to whether more robust intergroup differences were obscured by considerable biological variation in this human cohort. To test this, we employed a leave-one-out cross-validated (LOOCV) random forest machine-learning model, designed to identify distinguishing data features from complex, high-dimensional data (15). The recursive feature selection approach we adopted chose differing combinations of study time points as model inputs, which were scaled to the first baseline time point (Fig. 4A).

To gain insight into whether microbial or host protein abundance on specific days was more predictive in classifying these individuals based on their augmented diets, we ran the classifier only on host or on microbial proteins, considering either individual

**FIG 3** Legend (Continued)

proteome. Colors overlaid on each node are the result of Markov-clustering algorithm employed by StringDB. (D) Universally identified proteins significantly ($P < 0.05$) altered during the final day of the maintenance period between fermented and fiber group diets. Proteins were normalized, $\log_2$ transformed, and scaled to levels present on the first baseline day ($\log_{2day} - \log_{2baseline}$). Proteins were also filtered for variation between participants ($\sigma/\sigma_{max\ variation} > 0.15$).

time points (e.g., only ramp, only maintenance day 1, etc.) or aggregated time points based on intervention status (e.g., all three postdiet intervention days) (Fig. 4A). Overall, we found that the greatest classification accuracy was achieved by considering the abundance of solely host proteins from the final maintenance measurement (89%). In contrast, microbial proteins measured at this time point only yielded 78% accuracy. However, it is noteworthy that evaluating the ramp and both maintenance time points together improved this classification somewhat for microbial proteins (80%) but decreased classification accuracy for host proteins (72%). These data suggest a more comprehensive microbial profile measured following diet induction captured both transient and sustained diet-specific signals, whereas host protein expression tended to evolve over the course of the intervention.

Given that host proteins better predicted group membership (Fig. 4A), we next varied the number of participants included in the model (Fig. 4B) to test whether this observation depended on the underlying depth of the data set. As expected, we observed increased classification accuracy as more study participants were included in the model (averages of 51%, 65%, and 80% accuracy for 10, 20, and 28 participants, respectively), signifying the necessity of more extensive data sets for studies focused on disease prediction and explanatory power. Despite being less than 10% of each sample's proteomic profile, these data support host proteins' ability to generate a more accurate classifier than microbial proteins.

## DISCUSSION

Stool-based proteomics' potential use as a basic science tool and a rich resource for clinical biomarker discovery has been touted for over a decade (8, 16, 17). However, its broad adoption has been largely hindered by multiple difficulties. Chief among them is processing large sample numbers in an efficient manner. Achieving this is a necessary prerequisite for undertaking studies involving large heterogenous populations, such as human trials. While some protein extraction and digestion protocols were recently designed to have robust sample processing pipelines, their targeted throughput of 5 to 25 samples per week makes scaling them with automation difficult. This limits their usefulness to large, longitudinal clinical studies. Relatedly, several protocols we evaluated prior to developing SHT-Pro failed to remove major contaminating molecules found in stool, which was evident from the continuous fouling of liquid chromatography columns and unstable mass spectrometer performance (7). Both necessitate increased equipment maintenance and associated downtime (7). SHT-Pro resolves these deficiencies by leveraging a workflow specifically designed for large longitudinal stool collections while maintaining flexibility to accommodate smaller sample numbers for pilot studies. Importantly, this is accomplished with a high degree of experimental reproducibility. Our previously published methods required the processing of large sample sets over multiple months, leading to a greater need to distinguish prominent preparation artifacts from desired biological protein profiles. Here, we show that SHT-Pro can produce highly reproducible data sets spanning hundreds of samples in a matter of days. For example, in the current study, SHT-Pro saved an estimated 3.5 months (approximately 80% less time) over our previous protocol. Since it is compatible with multiplexing technology, LC-MS data generation times can be further accelerated by an additional order of magnitude. Similar to the high-throughput DNA sequencing pipelines used to characterize gut microbial communities on a massive scale, we envision SHT-Pro will modernize the stool proteomics field and allow the profiling of a variety of diseased conditions, ranging from IBD to multiple sclerosis (18). Future studies may also consider adapting high-throughput proteome preparation pipelines, such as 96-well FASP (19), or all-in-one commercial kits, such as PreOmics' iST kit and ThermoFisher Scientific's Easypep kits. We note, however, that in our unpublished pilot studies, methods that perform quite well with cell or tissue lysate tended to be overwhelmed by stool's molecular diversity (data not shown).

Nevertheless, we acknowledge that SHT-Pro as described here could be further improved in several simple ways. First, the aliquoting of initial stool samples is presently

the most labor-intensive and time-consuming step of the overall process. In our diet study, each sample was aliquoted by hand from the original specimen collection vessel to the 96-well bead-beating plate (Fig. 1A). Given that this is a common obstacle for DNA and protein sample preparation pipelines alike, the microbiome field would benefit from an aliquoting technology targeted at this sample-handling burden. Next, while multichannel pipettes currently used in SHT-Pro were critical components, a 96-well pipettor (single head or as part of an automated system) could more uniformly and rapidly dispense buffers, thereby increasing the overall speed of the assay while decreasing the amount of hands-on time laboratory researchers must invest in an experiment and decreasing preparative variation. Additionally, we noted a large portion of time within SHT-Pro was spent evaporating and concentrating samples via Cetrivap/Speedvac vacuum-based concentrators (Fig. 2B). Given the larger volumes 96-well plates produce using our method (100 to 300 $\mu$l/well), this can be a significant hindrance to overall throughput. As such, an alternative method to concentrate peptides would significantly increase the throughput of SHT-Pro, potentially bringing sample preparation time to less than a day. Last is the issue of cost, which can become a deciding factor when selecting a protocol. Not including automation hardware, SHT-Pro can cost up to approximately $30 per sample when using the 96-well plate method, which is substantially more than a common in-solution digest. However, this must be weighed against the additional time and manpower it takes to process those same samples over a substantially longer time period.

In the current study, we observed over 5,600 proteins that could provide new biological insights into the impact of dietary fiber versus fermented foods. While this identification depth is greater than some of the first reported metaproteomic searches on stool, newer studies have reported substantially more host and microbial protein identifications (53,000 total proteins, various biological matrices) (9, 16, 20). We attribute the decreased number of identifications in our current study to several factors. First, our use of the TMT multiplexing reagent creates a bias toward proteins that are found in multiple samples: signals found in just one sample are diluted by the number of channels used (21). Thus, we suspect many low-abundant, sample-specific host and microbial proteins were not identified. To combat this, future iterations of SHT-Pro could incorporate peptide fractionation and longer mass spectrometry runs per sample, which has been shown to significantly increase identification of low-abundance host and microbial proteins (9). In this context, striking a balance between throughput and proteomic depth is crucial, as the biological and health-related significance of low-abundance proteins remains promising but unclear. Next, while the database used to search these samples (adapted from the Human Microbiome Project) is fairly extensive, the use of subject-specific metagenomes for the generation of protein databases would likely increase sample- and subject-specific protein identifications. Lastly, compared to previously published work, we injected approximately 4× less material into the mass spectrometer (0.5 $\mu$g versus 2 $\mu$g) (20). Given that, on average, we collected approximately 60 $\mu$g of peptide from each sample (over 100 $\mu$g/sample was collected in the pilot study), injecting more peptide or fractionating samples would likely increase our protein identification rate.

Despite these remaining challenges, SHT-Pro-generated metaproteome data that resulted in biologically meaningful insights, even in the context of a largely uncontrolled human diet study. Indeed, SHT-Pro revealed a subtle divergence in proteomes after the introduction of fiber and fermented diets, as evidenced by the increased number of significantly altered host and microbial proteins during the ramp and maintenance phases, while baseline measurements remained largely unchanged. These significantly increased proteins were enriched for several categories, including intestinal epithelium maintenance and host sphingolipid metabolism. Interestingly, sphingolipids, along with chemical variants (e.g., glycosphingolipids) and derivatives, previously were shown to regulate invariant natural killer T cells (iNKT) (22). More recently, *Bacteroides fragilis*, a common gut-dwelling microbe, has been shown to produce sphingolipids, and their production protected mice from an oxazolone-induced colitis

model, an effect largely mediated by their regulation of iNKT activation (23). Here, the introduction of fermented foods may have increased the levels of *B. fragilis*, as has been previously noted in rats fed fermented tempeh, which in turn may increase levels of sphingolipid availability (24). In the current study, we observed 20 proteins attributed to *B. fragilis*; however, they showed no significant abundance differences between diet cohorts on the final day of maintenance. It is possible that the search algorithm used (TurboSequest, Proteome Discoverer 2.2) was not ideally suited to attribute peptides (and proteins) to the correct species in such a large search space, a common problem in the metaproteomic field (3). In this case, metaproteome-centric search suites, such as MetaLab, may be of some benefit (25). While the purpose of the manuscript is to showcase SHT-Pro as an integral facet necessary for understanding host-microbe interactions, this result suggests that future studies using SHT-Pro would also benefit from a multiomic approach that also leverages 16S rRNA amplicon sequencing and metabolomics profiling. Nevertheless, SHT-Pro-generated data are compelling when considering that, other than dictating increased intake of each experimental cohort's respective diet, study participants had no other nutritional restrictions. As such, any changes in the microbial or host stool proteome could be expected to be subtle and subject specific and likely hidden by data-driven noise. This subtlety is highlighted by the classification success, which was only possible using machine-learning techniques and not easily discernible by simply focusing on simple abundance changes. Importantly, the observed success of the LOOCV random forest model also suggests future microbial proteomic studies would benefit from normalization to a participants' unique baseline signature as well as the inclusion of many participants, an inherent strength of SHT-Pro.

These data likely harbor many more insights, including revealing components of diet. While we have not mapped dietary peptides in this study due to database limitations, plant peptides are evident in our data set and suggest utility in helping inform the many challenging aspects of dietary assessment in free-living humans. When paired with other omic data (e.g., 16S rRNA, metabolomics, and clinical measurements), these proteomic profiles are poised to significantly contribute to our understanding of the dietary impact on individuals over time. Questions such as these, which require much more in-depth analysis, will be answered in a separate manuscript, and the focus of this article is largely SHT-Pro's increase in quality and speed compared to the previous workflow, and more in-depth analysis of multiomic data associated with the dietary intervention will be completed as part of a larger publication.

Nevertheless, taken together, SHT-Pro reveals itself as a robust pipeline for processing stool samples in an extremely timely manner, and we believe its wide-scale adoption and improvement will enable powerful discoveries in the field of host-gut microbiome interactions.

## MATERIALS AND METHODS

**Buffers.** For the lysis buffer, 6 M urea, 5% sodium dodecyl sulfate (SDS), and 50 mM Tris were combined, with the pH adjusted to 8 using phosphoric acid. Roche cOmplete Mini protease inhibitor cocktail (04693159001; Roche) was added prior to adding buffer to samples. The Protifi binding buffer (PBB) contained 90% methanol and 10% triethylammonium bicarbonate buffer (TEAB; catalog number T7408; Sigma-Aldrich), adjusted to pH 7.1 using phosphoric acid. The digestion buffer contained 100 mM TEAB and 5 $\mu$g trypsin (V5113; Promega). For peptide elution buffers, the first elution was performed using digestion buffer, the second elution was performed using 0.2% formic acid (FA), and the third elution was performed using 50% acetonitrile and 0.2% FA.

**Isolation of stool proteins and peptides (96-well variant).** A step-by-step guide is available on Protocols.io at the following web address: https://doi.org/10.17504/protocols.io.9gph3vn. Approximately 100 to 200 mg (when available) from each collected stool specimens was aliquoted into a 96-well plate along with approximately 600 mg of 0.1-mm ceramic beads (27-6006; Omni International). To each filled well, 750 $\mu$l of lysis buffer was added and plates were sealed with the Omni-provided sealing mats. To increase their seal, each plate was additionally sealed with parafilm, although we found this was not necessary. The sealed plates were subjected to 10 min of bead beating at 20 Hz using a Qiagen TissueLyser II. After bead beating, each plate was centrifuged at 300 relative centrifugal force (RCF) at 4°C for 10 min. Five hundred microliters of the resulting supernatant was transferred to a new 2-ml 96-well plate (186002482; Waters), sealed with a sealing mat, spun again at 300 RCF at 4°C for 10 min, and then transferred into a fresh 2-ml plate. Samples then were reduced with 10 $\mu$l of 50 mM dithiothreitol

(Sigma-Aldrich) for 30 min at 47°C and alkylated with 30 $\mu$l of 50 mM iodoacetamide (Sigma-Aldrich) for 1 h at room temperature in the dark. Fifty microliters of the reduced and alkylated supernatant was transferred to a new 2-ml 96-well plate for further processing, while the remaining material was stored at −80°C for potential future analysis. Supernatant-resident stool proteins were washed, digested, and eluted as described in the Protifi S-trap protocol (see http://www.protifi.com/wp-content/uploads/2018/08/S-Trap-96-well-plate-long-1.4.pdf for the complete protocol). Briefly, 50 $\mu$l of supernatant was acidified with 5 $\mu$l of 12% phosphoric acid, to which 300 $\mu$l of S-trap binding buffer was added. Each resulting mixture was loaded into a single well. Positive pressure was used to load the proteins into each well (Waters Positive Pressure-96 processor) with pressure at approximately 6 to 9 lb/in$^2$ on "low-flow" setting. Note that if, after 1 min, volume still remains in the well, using a pipette tip to move any debris to the side of well will begin the flow again. Loaded proteins were washed with 300 $\mu$l PBB five times. After washing, 125 $\mu$l of digestion buffer was added and proteins were digested for 3 h at 47°C. Peptides were then eluted with 100 $\mu$l TEAB, followed by 100 $\mu$l of 0.2% formic acid, followed by 100 $\mu$l of 50% acetonitrile (ACN), 0.2% formic acid. These were captured in a 1-ml 96-well plate (AB-1127; Thermo Scientific), and the volume was dried down in a Centrivap SpeedVac (model 7810016). Plated samples then were desalted using RP-S tips on the Agilent Bravo AssayMap using a built-in desalting protocol, eluted with 50% ACN, and dried down. Plated peptide concentration was normalized using readings from the Biotek Synergy microplate reader and the Take3 microvolume plate (single samples were adjusted using a NanoDrop ND-1000). Samples then were labeled with a TMT-11 multiplexing kit using the manufacturer's recommended method (A34808; Thermo-Fisher Scientific). Channel-specific isobaric tag intensities were adjusted to 11 (1:1) using recorded intensities from a 1-h gradient mass spectrometry run and subsequently reinjected into the mass spectrometer after normalization.

**Isolation of stool proteins and peptides (individual tube variant).** Approximately 100 to 200 mg (when available) from each collected stool specimens was aliquoted into a bead-beating tube along with approximately 600 mg of 0.1-mm ceramic beads (19-732; Omni International). To each tube, 750 $\mu$l of lysis buffer was added. Samples were subjected to 10 min of bead beating at 3,500 rpm (Omni Beadruptor 12 19-050). After bead beating, each sample was centrifuged at 300 RCF at 4°C for 10 min. Five hundred microliters of the resulting supernatant was transferred to a fresh 2-ml tube, spun again at 300 RCF at 4°C for 10 min, and then again transferred to a fresh 2-ml plate. Samples then were reduced with 10 $\mu$l of 50 mM dithiothreitol (Sigma-Aldrich) for 30 min at 47°C and alkylated with 30 $\mu$l of 50 mM iodoacetamide (Sigma-Aldrich) for 1 h at room temperature in the dark. Fifty microliters of the reduced and alkylated supernatant was transferred to a new 2-ml tube for further processing, while the remaining material was stored at −80°C for potential future analysis. Supernatant-resident stool proteins were washed, digested, and eluted as described in the Protifi S-trap protocol (see http://www.protifi.com/wp-content/uploads/2018/08/S-Trap-mini-protocol-long.3.6.pdf for the complete protocol). Briefly, 50 $\mu$l of supernatant was acidified with 5 $\mu$l of 12% phosphoric acid, to which 300 $\mu$l of S-trap binding buffer was added. Each resulting mixture was loaded into a single well. A vacuum manifold was used to load samples with pressure set at approximately 3 to 5 lb/in$^2$. Note that if, after 1 min, volume still remains in the well, using a pipette tip to move any debris to the side of the well will begin the flow again. Loaded proteins were washed with 300 $\mu$l PBB five times. After washing, 125 $\mu$l of digestion buffer was added and proteins were digested for 3 h at 47°C. Peptides were then eluted with 100 $\mu$l TEAB, followed by 100 $\mu$l of 0.2% formic acid, followed by 100 $\mu$l of 50% acetonitrile, 0.2% FA. Eluate was captured and the volume was dried down in a Centrivap SpeedVac (model 7810016). Dried samples were then resuspended in 250 $\mu$l 0.2% FA. Resuspended samples were then desalted using Seppak tC$_{18}$ cartridges and subsequently dried down (WAT036820; Waters). Each sample was then resuspended in 30 $\mu$l and the peptide concentration was normalized (NanoDrop ND-1000).

**Previous workflow protocol.** Samples were prepared as described in Gonzalez et al. (6). Briefly, sample pellets were disrupted using 500 $\mu$l 8 M urea lysis buffer supplemented with Roche cOmplete protease inhibitor (04693159001; Roche) by vortexing. After pellet resuspension, insoluble material was pelleted down at 2,500 RCF for 10 min at 4°C, and the collected supernatant was subjected to ultracentrifugation (35,000 rpm for 30 min at 4°C; Beckman-Coulter Optima Ultracentrifuge) to remove bacteria. The ultracentrifuge supernatant was subsequently reduced, alkylated, and precipitated overnight in a −20°C freezer using trichloroacetic acid (15% total volume). Protein pellets were resuspended in 40 $\mu$l of loading buffer and briefly run in SDS-PAGE (approximately 5 mm; Invitrogen NuPAGE 4 to 12% Bis-Tris) for further purification, after which they were subjected to in-gel tryptic digestion using sequencing-grade trypsin (V5113; Promega). After digestion, each sample was cleaned up using C$_{18}$ columns and dried down. Peptides were then normalized using a NanoDrop ND-1000.

**Mass spectrometry.** Peptide samples were diluted to 0.5 $\mu$g/$\mu$l. Subsequently, 1 $\mu$l was loaded onto an in-house laser-pulled 100-$\mu$m-inner-diameter nanospray column packed to ∼220 mm with 3-$\mu$m 2Å C$_{18}$ beads (Reprosil). Peptides were separated by reverse-phase chromatography on a Dionex Ultimate 3000 high-performance liquid chromatograph (HPLC). Buffer A of the mobile phase contained 0.1% FA in HPLC-grade water, while buffer B contained 0.1% FA in ACN. An initial 2-min isocratic gradient flowing 3% B was followed by a linear increase up to 25% B for 115 min, increased to 45% B over 15 min, and a final increase to 95% B over 15 min, whereupon B was held for 6 min and returned to baseline (2 min) and held for 10 min, for a total of 183 min. The HPLC flow rate was 0.400 $\mu$l/min. Samples were run on either a Thermo Fusion Lumos (large study) or Thermo Orbitrap Elite (pilot comparisons) mass spectrometer that collected MS data in positive ion mode within the 400 to 1,500 $m/z$ range.

For TMT-labeled samples, a top-speed MS3 method was employed on the Fusion Lumos with an initial Orbitrap scan resolution of 120,000. This was followed by high-energy collision-induced dissociation and analysis in the Orbitrap using "Top Speed" dynamic identification with dynamic exclusion enabled (repeat

count of 1, exclusion duration of 90 s). The automatic gain control for Fourier transform (FT) full MS was set to 4e5 and for ITMSn was set to 1e4. ITCID was used with the MS2 method, and the MS3 AGC was set to 1e5.

**Peptide/protein searches.** The resulting mass spectra raw files were first searched using Proteome Discoverer 2.2 using the built-in SEQUEST search algorithm. Built-in TMT batch correction was enabled for all samples. Three FASTA databases were employed: Uniprot Swiss-Prot *Homo sapiens* (taxon ID 10090, downloaded January 2017), the Human Microbiome Project database (FASTA file downloaded from https://www.hmpdacc.org/hmp/HMRGD/ in January 2017), and a database containing common sample-handling contaminants. Target-decoy searching at both the peptide and protein level was employed with a strict FDR cutoff of 0.05 using the Percolator algorithm built into Proteome Discoverer 2.2. Enzyme specificity was set to full tryptic with static peptide modifications set to carbamidomethylation ($+57.0214$ Da) and, when appropriate, TMT ($+229.1629$ Da). Dynamic modifications were set to oxidation ($+15.995$ Da) and N-terminal protein acetylation ($+42.011$ Da). Only high-confidence proteins ($q < 0.01$) were used for analysis.

**Statistical analyses.** Statistics were calculated using R with statistics packages (FactoMinerR 1.36, factoextra 1.0.5, ggplot2 2.2.1, Hmisc 4.0-3, psych 1.7.8, Mfuzz 2.34.0, ggpubr 0.1.5, RColorBrewer 1.1-2, UpSetR, 1.3.3, limma 3.30.13, and venneuler 1.1-0) and Qlucore Omics Explorer 3.3. Protein abundance was normalized as a percentage of summed reporter intensity for all quantified proteins in a given sample (protein intensity/total sample intensity). Each TMT-11 run was filtered for. Where necessary for meeting statistical assumptions, abundances were $\log_2$ transformed. The appropriate multiple-hypothesis tests (one-way analysis of variance) were applied to abundance comparison data using Qlucore Omics Explorer or custom R scripts. Correlational $P$ values were corrected using the FDR setting and the R package psych 1.7.8. Protein abundance heat maps were generated with Qlucore Omics Explorer 3.3 or R's built-in heatmap function. FDRs and fold changes (where appropriate) were generated using Qlucore's built-in FDR estimator, and the values are reported in tables in the supplemental material.

**Data availability.** The mass spectrometry proteomics data have been deposited to the ProteomeXchange Consortium via the PRIDE partner repository with the data set identifier PXD017450.

## SUPPLEMENTAL MATERIAL

Supplemental material is available online only.

**FIG S1**, JPG file, 0.7 MB.
**FIG S2**, JPG file, 0.4 MB.
**FIG S3**, JPG file, 0.7 MB.
**FIG S4**, JPG file, 1.7 MB.
**FIG S5**, JPG file, 1.8 MB.
**FIG S6**, JPG file, 2.1 MB.
**TABLE S1**, XLSX file, 1.5 MB.
**TABLE S2**, CSV file, 14.5 MB.
**TABLE S3**, CSV file, 0.2 MB.
**TABLE S4**, CSV file, 0.002 MB.

## ACKNOWLEDGMENTS

We thank members of the Elias and Sonnenburg laboratories for their valuable input during the experimental and manuscript writing phases.

We acknowledge the following sources of funding: C.G.G., HHMI Gilliam Fellowship, Stanford Graduate Fellowship in Science and Engineering; C.G.G. and H.C.W., NSF Graduate Fellowship (DGE–114747); J.E.E., Precision Health and Integrated Diagnostic (PHIND) Center at Stanford and the Chan Zuckerberg Biohub; J.L.S., NIH grants DK085025 and AT00989203. J.L.S. is a Chan Zuckerberg Biohub Investigator.

We have no conflicts of interest to report.

The SHT-Pro pipeline was designed by C.G.G. Proteomic sample preparation and mass spectrometry were conducted by C.G.G., H.C.W, and M.T. Samples were obtained by H.C.W., M.T., and J.L.S. Statistical analysis was done by C.G.G. and H.C.W. LOOCV random forest was generated by H.C.W. Manuscript writing and figure generation was done by C.G.G. and H.C.W., and editing was done by C.G.G., H.C.W., M.T., J.L.S., and J.E.E.

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
