## [Reviewer comments · mSystems]

High-Throughput Stool Metaproteomics: Method and Application to Human Specimens

Carlos Gonzalez, Hannah Wastyk, Madeline Topf, Christopher Gardner, Justin Sonnenburg, and Joshua Elias

Corresponding Author(s): Joshua Elias, Chan Zuckerberg Biohub

Review Timeline:

Submission Date:	March 5, 2020
Editorial Decision:	April 9, 2020
Revision Received:	May 14, 2020
Accepted:	June 2, 2020

Editor: Pieter Dorrestein

Reviewer(s): The reviewers have opted to remain anonymous.

Transaction Report:

DOI: <https://doi.org/10.1128/mSystems.00200-20>

April 9, 2020

Dr. Joshua E Elias
Chan Zuckerberg Biohub
Mass Spectrometry Platform
318 Campus Drive
Clark Center, Room W300C
Stanford, CA 94305

Re: mSystems00200-20 (High-Throughput Stool Metaproteomics: Method and Application to Human Specimens)

Dear Dr. Joshua E Elias:

I hope all is well during these trying times. The paper needs minor revisions.

Below you will find the comments of the reviewers.

To submit your modified manuscript, log onto the eJP submission site at <https://msystems.msubmit.net/cgi-bin/main.plex>. If you cannot remember your password, click the "Can't remember your password?" link and follow the instructions on the screen. Go to Author Tasks and click the appropriate manuscript title to begin the resubmission process. The information that you entered when you first submitted the paper will be displayed. Please update the information as necessary. Provide (1) point-by-point responses to the issues raised by the reviewers as file type "Response to Reviewers," not in your cover letter, and (2) a PDF file that indicates the changes from the original submission (by highlighting or underlining the changes) as file type "Marked Up Manuscript - For Review Only."

Due to the SARS-CoV-2 pandemic, our typical 60 day deadline for revisions will not be applied. I hope that you will be able to submit a revised manuscript soon, but want to reassure you that the journal will be flexible in terms of timing, particularly if experimental revisions are needed. When you are ready to resubmit, please know that our staff and Editors are working remotely and handling submissions without delay. If you do not wish to modify the manuscript and prefer to submit it to another journal, please notify me of your decision immediately so that the manuscript may be formally withdrawn from consideration by mSystems.

To avoid unnecessary delay in publication should your modified manuscript be accepted, it is important that all elements you upload meet the technical requirements for production. I strongly recommend that you check your digital images using the Rapid Inspector tool at <http://rapidinspector.cadmus.com/RapidInspector/zmw/>.

Corresponding authors may join or renew ASM membership to obtain discounts on publication fees.

Need to upgrade your membership level? Please contact Customer Service at Service@asmusa.org.

Sincerely,

Pieter Dorrestein

Editor, mSystems

Journals Department
Reviewer comments:

Reviewer #1 (Comments for the Author):

In this manuscript, the authors devised a high throughput fecal metaproteomic sample preparation procedure, termed SHT-Pro, based on commercially available 96-well S-trap column and TMT labeling strategy. This approach was then applied in a dietary intervention study consisting of 145 stool samples (in duplicates), which revealed the responses of fecal human and/or microbial proteins to either fermented- or fiber-enriched diet in human volunteers. The study was well designed, and the approach is of interest to the field and will benefit greatly the application of metaproteomics in large scale microbiome studies. There are several minor concerns need to be addressed as indicated below.

(1) SHT-Pro is an S-trap column-based protein purification/digestion workflow (a type of in-column digestion), however in this study the authors compared SHT-Pro with their previous in-gel digestion workflow. While this comparison is fine, it will be much more meaningful to compare SHT-Pro with other in-column digestion approaches, such as FASP. In-solution protein digestion is among the most widely used approaches in both metaproteomic and typical proteomic studies. The authors may want to discuss the advantages and disadvantages of different approaches in the context of microbiome studies in such as a methodology study. In addition, is the cost a side effect?

(2) One challenge of the fecal metaproteomics is the large variation in the consistency of stool samples. There could also be undigested food debris in some stool samples. Altogether these variations will dramatically affect the yield of proteins. How did the authors normalize these variations? One potential detrimental effect related to this variation in SHT-Pro approach might be the different trypsin-to-protein ratios during protein digestion. To what extent would the trypsin-to-protein ratios be affected, and to what extent would these different ratios influence the eventually metaproteome profiles?

(3) In dietary intervention study, the host proteins were shown to better predict group membership compared to microbial proteins. Are functional profiles of microbiome more predictive for the groups? Different individuals may have different microbial composition and thereby different protein

membership depending on how the database was compiled. Assigning different proteins to functional orthologues may potentially address these biases given the known high functional redundancy of human gut microbial species.

Others:

(4) Line 111: is ref10 relevant?

(5) Lines 122-130: this paragraph is very difficult to understand. In addition, it seems the subject discussed in this paragraph is not very relevant to SHT-Pro. The MS contamination depends on many factors, such as loading amount, instrument types, etc. It might be much more dependent on the performance of desalting, which is not the unique part of SHT-Pro workflow.

(6) Line 141: did the author recommend a minimum of 100mg starting material as the R2 value of 50 mg is much lower than both 100 mg and 200 mg?

(7) Line 182: Whether the top 100 proteins were mainly host proteins? It will be interesting to see the functions of top100 microbial proteins as well.

(8) Figure 2G: the legend didn't match the figure.

(9) Lines 279-282: there is no way to see subject-specific clustering from the figures in SI6. It is needed to either change the color coding or perform statistical analysis for inter-individual variations.

Reviewer #2 (Comments for the Author):

Happy to provide feedback on the SHT-Pro paper for metaproteomics sample processing. The authors detail a new method for sample prep that reduces the time spent processing stool samples and helps increase the number of samples that can be processed within a unit time. I have several minor suggestions listed below

- the introduction claims a 100-fold time savings, while elsewhere in the manuscript a 10-fold improvement is claimed. I think this is just a typo.

- line 122, second results paragraph starting 'Sample processing improvements would be of little value...'. The claim here about data quality is in relation to 'contaminant removal'. However the rest of the paragraph does not mention contaminants at all. Nor does the result compare with any older method to show improvement. It merely says that a interspersed peptide standard looks stable. Please either show comparison with your oft cited previous method, or talk about contaminants, or both.

- Kind of surprised that the improvement is only 10x given that you are 10x multiplexing with TMT. What advantage was the rest of the sample prep?

- I guess related to that the time savings as shown in the figures is only for sample prep, but not data acquisition. By TMT multiplexing you presumably save on instrument time as well. Does that merit inclusion and comparison?

- very excited to see that the things most useful in sample classification were host proteins. It helps the study, which is supposed to see if diet improves the host. Thought this part was under-emphasized. But perhaps you have another paper talking about that.

Reviewer #1 main comments:

1. SHT-Pro is an S-trap column-based protein purification/digestion workflow (a type of in-column digestion), however in this study the authors compared SHT-Pro with their previous in-gel digestion workflow. While this comparison is fine, it will be much more meaningful to compare SHT-Pro with other in-column digestion approaches, such as FASP. In-solution protein digestion is among the most widely used approaches in both metaproteomic and typical proteomic studies.
 - a. The authors may want to discuss the advantages and disadvantages of different approaches in the context of microbiome studies in such as a methodology study.
- We thank the referee for bringing this excellent point up. We believe the comparison of FASP, as well as the newer commercial products such as PreOmics' iST and Thermo EasyPep kits could be useful. However, while we have yet to test iST or EasyPep kits with stool, we previously piloted FASP (non-high-throughput version) and found the stool matrix overwhelmed the size filtration apparatus, causing it to clog and contaminate the digested peptide eluate with visible bile salts and lipids. This occurred even with less starting material than we described in this study. With long filtering times, unacceptably contaminated samples and surprisingly few protein identifications compared to our previous in-solution or in-gel methods, we abandoned the FASP approach early on, and did not see the value in adopting previously published versions, as referenced below. While it remains unclear how FASP might quantitatively compare to SHT-Pro, our pilot studies were sufficient for us to determine that our efforts were best focused on other approaches.
 - Yu Y, Bekele S, Pieper R. Quick 96FASP for high throughput quantitative proteome analysis. *J Proteomics*. 2017;166:1–7
- In-solution digest was part of our first fecal proteome method (see Lichtman et. al 2013). We transitioned from this protocol to the limited in-gel digest protocol due to heavy sample, column, and mass spec contamination and unacceptable reproducibility. The addition of the in-gel digest (See references below) mitigated some contamination but did not remove it sufficiently. Moreover, this process was too manually intensive for us to commit very large sample sets to it.
- We have further added the following text to the manuscript to address these two points:
 - Line 351 – “Future studies may also consider adapting high-throughput proteome preparation pipelines such as 96-well FASP (20) or all-in-one commercial kits such as PreOmics' iST kit and ThermoFisher Scientific's Easypep kits. We note, however, that in our unpublished pilot studies, methods which perform quite well with cell or tissue lysate tended to be overwhelmed by stool's molecular diversity.”
 - References:

- Tropini C, Moss EL, Merrill BD, et al. Transient Osmotic Perturbation Causes Long-Term Alteration to the Gut Microbiota. *Cell*. 2018;173(7):1742-1754.e17. doi:10.1016/j.cell.2018.05.008
- Casavant E, Park KT, Elias JE. Proteomic Discovery of Stool Protein Biomarkers for Distinguishing Pediatric Inflammatory Bowel Disease Flares [published online ahead of print, 2019 Sep 6]. *Clin Gastroenterol Hepatol*. 2019;S1542-3565(19)30974-7. doi:10.1016/j.cgh.2019.08.052

b. In addition, is the cost a side effect?

- Considering employee salaries, and the value of generating data sets from 100's rather than dozens of specimens, the cost of ~\$30 per specimen (excluding LC-MS instrumentation) is quite low and worth the expense.
 - We have further added the following text to the manuscript to address this:
 - Line 374 – “Last is the issue of cost, which can become a deciding factor when selecting a protocol. Not including automation hardware, the SHT-Pro can cost up to approximately \$30 per sample when using the 96 well plate method, which is substantially more than a common in-solution digest. However, this must be weighed against the additional time and manpower it takes to process those same samples over a substantially longer time period.”
 - We now also reference our protocol on the protocols.io platform (<https://dx.doi.org/10.17504/protocols.io.9gph3vn>) that includes all consumable part numbers, which should also help readers estimate their costs for implementing our procedure.
2. One challenge of the fecal metaproteomics is the large variation in the consistency of stool samples. There could also be undigested food debris in some stool samples. Altogether these variations will dramatically affect the yield of proteins.
- a. How did the authors normalize these variations?
- The reviewer brings up an excellent point. Due to the variations in initial weight as well as the unknown composition of each stool (e.g. proportion of food, water, microbes and host proteins), we choose to take an initial sample of approximately 200 mg (wet weight) and normalized at the peptide level further downstream – after trypsin digestion and initial c18 cleanup but prior to TMT labeling, as have other groups (see citation below).
 - Mills RH, Vázquez-Baeza Y, Zhu Q, et al. Evaluating Metagenomic Prediction of the Metaproteome in a 4.5-Year Study of a Patient with Crohn's Disease. *mSystems*.
 - We have underscored this important point

- Line 145 – “We note, however, we did not test the lower limit of initial starting material needed for SHT-Pro nor attempt to control for the large amount of variation found in stool sample consistency.”
 - b. One potential detrimental effect related to this variation in SHT-Pro approach might be the different trypsin-to-protein ratios during protein digestion. To what extent would the trypsin-to-protein ratios be affected, and to what extent would these different ratios influence the eventually metaproteome profiles?
 - The SHT-Pro pipeline attempts to account for variations in initial (and unknown) protein concentration by keeping a fairly high (estimated 1:10 to 1:20) trypsin to protein ratio. The S-trap has an advertised capacity of 100-200 ug of protein so regardless of how much protein is loaded, it is within the S-trap-suggested trypsin:protein ratio. Additionally, we increased digestion time from the S-trap’s suggested 1 hour to 3 hours at 47°C rather than 37 °C, which we have found minimized missed cleavages.
 - We have added the text:
 - Line 113 – “96-well protein trap columns (Protifi S-trap) are robust to a wide range of protein:trypsin ratios, making them suitable to stool specimens with varying protein content. Using them for initial protein purification and digestion along with automation technologies for solid-phase extraction cleanup are two critical components of this added efficiency.”
- 3. In dietary intervention study, the host proteins were shown to better predict group membership compared to microbial proteins.
 - a. Are functional profiles of microbiome more predictive for the groups?
 - This is a valid question and we acknowledge the abbreviated presentation of the data’s biological relevance and explanatory power. A majority of these questions will be answered in a forthcoming separate study which focuses on this and related questions. It will include several different types of omics data and extensive analyses combining all available information we and our collaborators have generated. We tried to limit the scope of the present manuscript to the method description, evaluation, and implementation.
 - b. Different individuals may have different microbial composition and thereby different protein membership depending on how the database was compiled. Assigning different proteins to functional orthologues may potential address these bias given the known high functional redundancy of human gut microbial species.
 - We thank the reviewer for their comments and we will incorporate them into the upcoming manuscript which will address these questions using orthogonal data sets in a more thorough manner.
- 4. Line 111: is ref10 relevant?
 - Thank you for pointing this out, it seems our reference software did not update the citations and this has been corrected.
 - Citation 6,10 is now 6,8. There were redundant citations to same paper.

5. Lines 122-130: this paragraph is very difficult to understand. In addition, it seems the subject discussed in this paragraph is not very relevant to SHT-Pro. The MS contamination depends on many factors, such as loading amount, instrument types, etc. It might be much more dependent on the performance of desalting, which is not the unique part of SHT-Pro workflow.
 - We apologize for the lack of clarity. We have re-written the paragraph:
 - Line 124 – “The sample processing speed improvements SHT-Pro provides would be of little value without effective contaminant removal. To evaluate the kind of contamination-dependent analytical degradation that can occur over time, we repeatedly injected a single SHT-Pro processed stool specimen into our mass spectrometer twenty times. Four analyses of a standard complex peptide mixture were interspersed throughout these stool LC-MS analyses: one prior to all stool LC-MS analysis, two spaced 10 stool analyses apart, one following 20 stool analyses). We observed no substantial degradation of LC-MS performance as measured from search results of the standard peptide mixture ($7,205 \pm 60$ unique peptides), versus four LC-MS analyses of the standard mixture on a new analytical column (average $7,350 \pm 150$ unique peptides). In contrast, we observed a 30% decrease in peptide spectral matches (PSMs) and 27% decrease in peptides identifications in our standard peptide mix using our previous method over a similar number of injections (n=16 injections, SI. 1). While sample purity and mass-spectrometer performance are also responsive to other factors such as desalting protocols, amount of sample loaded on to column and instrument type, these results suggest that peptides resulting from SHT-Pro pipeline are not substantially contaminated in a way that impairs sensitive LC-MS equipment.”
6. Line 141: did the author recommend a minimum of 100mg starting material as the R2 value of 50 mg is much lower than both 100 mg and 200 mg?
 - We did not state a recommended amount of starting material in the text, however, as the reviewer pointed out, for the best results we believe at least 100 mg starting material, as it likely samples the overall stool content to a higher degree than smaller amounts. Text has been added to clarify this.
 - Line 151 – “All input protein amounts produced strong linear correlations (R^2 values for 50 mg = 0.85, 100 mg = 0.92, 200 mg = 0.91; Fig. 1e), suggesting that approximately 100 mg of starting material is sufficient for technical reproducibility.”
7. Line 182: Whether the top 100 proteins were mainly host proteins? It will be interesting to see the functions of top100 microbial proteins as well.
 - We apologize for the brevity of the microbial functional analysis. This topic will be much more thoroughly in an upcoming paper.
8. Figure 2G: the legend didn't match the figure.
 - We thank the reviewer for pointing this out. We have corrected the legend.

- Text now reads: “**G**) Violin plot comparison of host protein intensity/microbial protein intensity using various scales. Data are Log_{10} transformed.”
9. Lines 279-282: there is no way to see subject-specific clustering from the figures in SI6. It is needed to either change the color coding or perform statistical analysis for inter-individual variations.
- We originally generated a graph and (now appended to SI. 7) to address this concern, however, we left this version out of the original submission due to the complexity of so many mapped factors being presented on a single graph.

- **SI 7. Proteins per individuals within each group** **A)** PCA of all microbial (5,372; left) or host proteins (307; right). Data used to generate these plots were normalized and log_2 transformed. Ovals were automatically drawn to capture >85 % of data points within each diet study group. **B)** Principle coordinate analysis (PCoA) generated using the same dataset used to generate SI 7A, but labeled by individual (color), timepoint (marker shape) and group (oval, drawn to capture >85% of data).

Reviewer #2 main comments

- the introduction claims a 100-fold time savings, while elsewhere in the manuscript a 10-fold improvement is claimed. I think this is just a typo.
 - We thank the author for their comments and can understand the confusion surrounding the two numbers. The time savings is dependent on whether TMT is used (~100x time savings) or not (~10x savings). We have added the following text to clarify this:

- Line 42 – “With it, a single researcher can process over one hundred stool samples per week for mass spectrometry analysis, approximately 10-100x faster than previous methods, depending on whether isobaric labeling is used or not.”
 - line 122, second results paragraph starting 'Sample processing improvements would be of little value...'. The claim here about data quality is in relation to 'contaminant removal'. However the rest of the paragraph does not mention contaminants at all. Nor does the result compare with any older method to show improvement. It merely says that a interspersed peptide standard looks stable. Please either show comparison with your oft cited previous method, or talk about contaminants, or both.
 - We thank the author for their comments, we have generated a new supplemental figure (SI. 1) to address this shortcoming and added the following text:
 - Line 133 - In contrast, we observed a 30% decrease in peptide spectral matches (PSMs) and 27% decrease in peptides identifications in our standard peptide mix using our previous method over a similar number of injections (n=16 injections, SI. 1).

- **“SI 1a.** We compared PSMs acquired from injections of our peptide standard before, during, and after injections of peptides purified using our previous pipeline (n=16 injections) and SHT-Pro (n=20 injections). To control for column and mass spectrometer variation, values are presented as a percentage of PSMs acquired from the standard prior to the first injection of either our previous method or SHT-Pro. Three randomly chosen, but sequentially acquired standards (interspersed approximately every 12 hours between a

variety of samples injected into our mass spectrometer) are included to compare stochasticity.”

- Kind of surprised that the improvement is only 10x given that you are 10x multiplexing with TMT. What advantage was the rest of the sample prep?
 - Forgive the confusion in time savings figures cited in text. As we note in our first response, to this reviewer, the 10x figure is with respect to previous methods and no TMT labeling, while the 100x figure mentioned (for instance on line 86) is with respect to TMT-labeled samples. The following text has been added:
 - Line 42 – “With it, a single researcher can process over one hundred stool samples per week for mass spectrometry analysis, approximately 10-100x faster than previous methods, depending on whether isobaric labeling is used or not.”
- I guess related to that the time savings as shown in the figures is only for sample prep, but not data acquisition. By TMT multiplexing you presumably save on instrument time as well. Does that merit inclusion and comparison?
 - We agree that TMT labeling did save instrument/acquisition time, however we opted to cite only prep time because downstream TMT labeling would likely be similar regardless of preparation pipeline.
- very excited to see that the things most useful in sample classification were host proteins. It helps the study, which is supposed to see if diet improves the host. Thought this part was under-emphasized. But perhaps you have another paper talking about that.
 - We apologize for the lack of in-depth biological analysis on the included data. As the reviewer intuited, the bulk of the data will be analyzed in a larger paper with several ‘omic’ datasets, while the focus of this paper was the SHT-Pro pipeline.

June 2, 2020

Dr. Joshua E Elias
Chan Zuckerberg Biohub
Mass Spectrometry Platform
318 Campus Drive
Clark Center, Room W300C
Stanford, CA 94305

Re: mSystems00200-20R1 (High-Throughput Stool Metaproteomics: Method and Application to Human Specimens)

Dear Dr. Joshua E Elias:

Hope all is well.

Your manuscript has been accepted, and I am forwarding it to the ASM Journals Department for publication. For your reference, ASM Journals' address is given below. Before it can be scheduled for publication, your manuscript will be checked by the mSystems senior production editor, Ellie Ghatineh, to make sure that all elements meet the technical requirements for publication. She will contact you if anything needs to be revised before copyediting and production can begin. Otherwise, you will be notified when your proofs are ready to be viewed.

Sincerely,

Pieter Dorrestein
Editor, mSystems

Journals Department
Fig S6: Accept

Fig S5: Accept

Fig S2: Accept

Fig S1: Accept

Fig S4: Accept

Table S2: Accept

Fig S3: Accept

Table S4: Accept

Table S1: Accept

Table S3: Accept